# Generalization of Neural Combinatorial Solvers Through the Lens of Adversarial Robustness

**Simon Geisler**[1*], **Johanna Sommer**[1*], **Jan Schuchardt**[1],
**Aleksandar Bojchevski**[2], **Stephan Günnemann**[1]
{`geisler, sommer, schuchaj, guennemann`}`@in.tum.de` | `bojchevski@cispa.de`
[1]Department of Informatics & Munich Data Science Institute, Technical University of Munich
[2]CISPA Helmholtz Center for Information Security

## Abstract

End-to-end (geometric) deep learning has seen first successes in approximating the solution of combinatorial optimization problems. However, generating data in the realm of NP-hard/-complete tasks brings practical and theoretical challenges, resulting in evaluation protocols that are too optimistic. Specifically, most datasets only capture a simpler subproblem and likely suffer from spurious features. We investigate these effects by studying adversarial robustness–a local generalization property–to reveal hard, model-specific instances and spurious features. For this purpose, we derive perturbation models for SAT and TSP. Unlike in other applications, where perturbation models are designed around subjective notions of imperceptibility, our perturbation models are *efficient and sound*, allowing us to determine the true label of perturbed samples without a solver. Surprisingly, with such perturbations, a sufficiently expressive neural solver does not suffer from the limitations of the accuracy-robustness trade-off common in supervised learning. Although such robust solvers exist, we show empirically that the assessed neural solvers do not generalize well w.r.t. small perturbations of the problem instance.

## 1 Introduction

Combinatorial Optimization covers some of the most studied computational problems. Well-known examples are the NP-complete SATisfiability problem for boolean statements or the Traveling Salesperson Problem (TSP). These problems can be solved efficiently with approximate solvers that have been crafted over the previous decades (Festa, 2014). As an alternative to engineered, application-specific heuristics, learning seems to be a good candidate (Bengio et al., 2021)

and was studied as a component in traditional solvers (e.g. Haim & Walsh (2009)). Despite deep learning for combinatorial optimization gaining attention recently, it is still an open question if and to what extent deep learning can effectively approximate NP-hard problems.

Moreover, there is a "Catch-22"; even if neural networks could solve NP-hard problems, generating the training data is either (a) *incomplete but efficient* or (b) *complete but inefficient* (or approximate). An *incomplete* data generator (a) crafts the problem instances s.t. their labels are known and a *complete* data generator (b) obtains the labels via (approximately) solving the random instances. Additionally, a *dense sample* is intractable even for moderate problem sizes due to the large problem space $\mathbb{X}$.

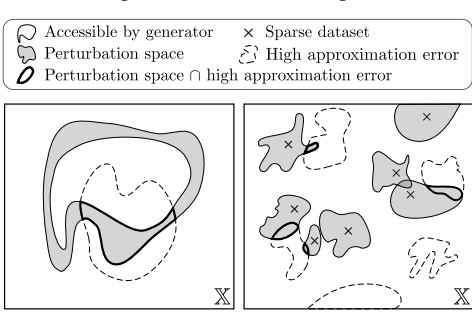

Figure 1: Adversarial examples can enhance the coverage of problem space $\mathbb{X}$ and find model-specific regions with difficult examples. Left, shows the asymptotic coverage of an efficient, incomplete data generator (Yehuda et al., 2020). Right, shows the important case ($\mathbb{X}$ is large) of a sparse sample of a possibly complete generator.

---

*equal contribution

For the special case of (a) exact polynomial-time data-generators, Yehuda et al. (2020) detail two challenges as a result of the NP-hardness: *(1) Easier subproblem:* a dataset from an efficient data generator only captures a strictly easier subproblem. *(2) Spurious features:* due to the lack of completeness, the resulting dataset can be trivially solvable due to superficial/spurious features. These findings extrapolate to case (b) for a sufficiently sparse sample (i.e. data could have been generated by an efficient data generator). Thus, it is concerning that often the same (potentially flawed) data generator is used for training *and evaluation*.

Adversarial robustness is a challenging local generalization property that offers a way of fixing the too optimistic model performance estimates that are the result of evaluations on incomplete datasets. We illustrate this in Fig. 1 for two relevant data generation schemes: (a) to the left we discuss an *incomplete, efficient* data generator and (b) to the right we discuss a sparse sample of a potentially *complete* data generator. With a suitable choice of perturbation model, we (a) possibly extend the reachable space by an efficient data generator or (b) cover the space around each instance in the sparse dataset. In other words, such adversarial attacks aim to find the intersection of the uncovered regions and hard, model-specific samples. Therefore, it is possible to detect the described defects. Adversarial robustness more realistically evaluates a model's generalization ability instead of simply testing on the *same* data generation procedure or a *sparse* external dataset.

Adversarial robustness is a desirable property for neural combinatorial optimization, since, in contrast to general learning tasks, in combinatorial optimization we do not have an accuracy robustness trade-off in the sense of Suggala et al. (2019). This means, there exists a model with high accuracy and high robustness. One key factor to avoid the accuracy robustness trade-off is choosing a perturbation model that guarantees the correct label of the perturbed sample (we call them *sound*). This is in stark contrast to other domains where one relies on imperceptible perturbations.

We instantiate our adversarial attack framework for the NP-complete SAT and TSP. Nevertheless, most of the principles can be transferred to other combinatorial optimization problems. Note that often such problems can be even reduced onto one another, as is the case for e.g. SAT and the maximum independent set. Having said this, we select SAT because of its general applicability and the notoriously challenging TSP due to its practical importance (e.g. for supply chain optimization). We then use our attacks to show that the evaluated neural SAT and TSP solvers are highly non-robust.

**Contributions.** **(1)** We bring the study of adversarial robustness to the field of neural combinatorial solvers to tackle fundamental problems in the evaluation of neural solvers. **(2)** We propose perturbation models for SAT and TSP s.t. we efficiently determine the updated solution. **(3)** We show that the models for SAT and TSP can be easily fooled with small perturbations of the problem instance and **(4)** that adversarial training can improve the robustness and generalization.

## 2 BACKGROUND ON NEURAL SOLVERS

Intuitively, combinatorial optimization is the task of finding an optimal element from the finite set of possible solutions (e.g. the truth assignment for a boolean statement). We formalize this as $Y = \arg\min_{Y' \in g(\boldsymbol{x})} c(\boldsymbol{x}, Y')$ where $\boldsymbol{x}$ is a problem instance, $g(\boldsymbol{x}) = \mathbb{Y}$ the finite set of feasible solutions, and $c(\cdot)$ a cost function. Typically, there is also an associated binary decision problem $y$, such as finding the optimal route vs. checking whether a route of at most cost $c_0$ exists (see § 4 and § 5). Then, for example, a neural solver $\hat{y} = f_\theta(\boldsymbol{x})$ learns a mapping $f_\theta : \mathbb{X} \to \{0, 1\}$ to *approximate* the decision problem. In this work, $\theta$ are the parameters, $\boldsymbol{x} \in \mathbb{X}$ is the problem instance, and $\hat{y}$ (or $\hat{Y}$) the prediction. In case of supervised learning, we then optimize the parameters $\theta$ w.r.t. a loss $\ell(f_\theta(\boldsymbol{x}), y)$ over a finite set of labeled training instances $(\boldsymbol{x}, y)$. However, to obtain the *exact* labels $y$ for a given $\boldsymbol{x}$ is intractable for larger problem instances due to the exponential or worse runtime. Two ways to generate data pairs are mentioned in the introduction and visualized in Fig. 1: (a) an efficient but incomplete data generator (b) using a solver to obtain the labels for random samples.

## 3 ADVERSARIAL ROBUSTNESS

Adversarial robustness refers to the robustness of a machine learning model to a small perturbation of the input instance (Szegedy et al., 2014). We define an adversarial attack in Eq. 1, where the parameters $\theta$ are constant and $G$ denotes the *perturbation model* that describes the possible perturbed

instances $\tilde{\boldsymbol{x}}$ around the clean sample $\boldsymbol{x}$ (i.e. the perturbation space) given the original solution $Y$. Since $\tilde{Y} \neq Y$ in the general case, we introduce $\tilde{Y} = h(\tilde{\boldsymbol{x}}, \boldsymbol{x}, Y)$ to model how the solution changes.

$$\ell_{\mathrm{adv},G}(\boldsymbol{x}, Y) = \max_{\tilde{\boldsymbol{x}}} \ell(f_\theta(\tilde{\boldsymbol{x}}), \tilde{Y}) \quad \text{s.t.} \quad \tilde{\boldsymbol{x}} \in G(\boldsymbol{x}, Y) \wedge \tilde{Y} = h(\tilde{\boldsymbol{x}}, \boldsymbol{x}, Y) \tag{1}$$

**Sound and efficient perturbation model.** Our framework for a neural combinatorial solver $f_\theta$ stands out from many other works on adversarial robustness since we choose the perturbation model $G$ s.t. we provably know a solution $\tilde{Y} = h(\tilde{\boldsymbol{x}}, \boldsymbol{x}, Y)$ for all possible $\tilde{\boldsymbol{x}}$. We call such a perturbation model *sound*. This stands in contrast to other domains, where we usually *hope to preserve* the label using the subjective concept of *imperceptible/unnoticable* perturbations (Szegedy et al., 2014).

While we can naively obtain a sound perturbation model for combinatorial optimization using a solver, this is intractable for realistic problem sizes. We therefore propose to use perturbation models that are *efficient and sound*. That is, we can determine the updated solution $\tilde{Y}$ without applying a solver on the perturbed instance $\tilde{\boldsymbol{x}}$. For example, if we add a node to a TSP instance, the optimal route including the new node will change, but we can efficiently determine $\tilde{Y}$ for the chosen $G$.

Important technical details arise due to (a) the potentially non-unique $Y$ and (b) non-constant $\tilde{Y}$ while perturbing the input. One way to handle both effects is through the choice of the loss $\ell$. (a) We can deal with the ambiguity in $Y$ if the loss is equal for any two optimal solutions/predictions. This can be achieved naturally by incorporating the cost $c(Y)$ of the combinatorial optimization problem. (b) Since the solution $\tilde{Y}$ can change throughout the optimization, it is important to choose a loss that assesses the difference between prediction $f_\theta(\tilde{\boldsymbol{x}})$ and ground truth $\tilde{Y}$. For example, a viable loss for TSP is the optimality gap $\ell_{\mathrm{OG}}(\hat{Y}, Y) = |c(\hat{Y}) - c(Y)| / c(Y))$ that is normalized by $c(Y)$.

**Perturbation strength.** With a sound perturbation model, all generated instances $\tilde{\boldsymbol{x}}$ are valid problem instances regardless of how much they differ from $\boldsymbol{x}$. Hence, in the context of combinatorial optimization, the perturbation strength/budget models the severity of a potential distribution shift between training data and test data. This again highlights the differences to other domains. For example in image classification with the common $L^p$ perturbation model $\|\boldsymbol{x} - \tilde{\boldsymbol{x}}\|_p \leq r$, the instance changes its true label or becomes meaningless (e.g. a gray image) for a large enough $r$.

**Generalization.** Specifically, adversarial robustness is one way to measure the generalization over perturbed instances $\tilde{\boldsymbol{x}}$ in the proximity of $\boldsymbol{x}$. Adversarial robustness is important in the context of neural combinatorial solvers since training and validation/test distribution differ from the actual data distribution $p(\boldsymbol{x})$. First, the data distribution $p(\boldsymbol{x})$ is typically unknown and highly application-specific. Second, due to theoretical limitations of the data generation process the train and validation/test distribution likely captures a simpler sub-problem suffering from spurious features (Yehuda et al., 2020). Third, we ultimately desire a general-purpose solver that performs well regardless of $p(\boldsymbol{x})$ (in the limits of a polynomial approximation).

We stress that in the context of combinatorial optimization, adversarial examples are neither anomalous nor statistical defects since all generated instances correspond to valid problem instances. In contrast to other domains, the set of valid problems is not just a low-dimensional manifold in a high-dimensional space. Thus, the so-called manifold hypothesis (Stutz et al., 2019) does not apply for combinatorial optimization. In summary, it is critical for neural solvers to perform well on adversarial examples when striving for generalization.

**Accuracy robustness trade-off.** A trade-off between adversarial robustness and standard generalization was reported for many learning tasks (Tsipras et al., 2019). That is, with increasing robustness the accuracy on the test data decreases. Interestingly, with a sound perturbation model and the purely deterministic labels in combinatorial optimization (the solution is either optimal or not), no such trade-off exists. Hence, if the model was expressive enough and we had sufficient compute, there would exist a model with high accuracy and robustness (see § A for more details).

**Adversarial training.** In adversarial training, we leverage adversarially perturbed instances with the desire of training a robust model with improved generalization. For this, adversarial attacks reveal the regions that are both difficult for the model and not covered by training samples (see Fig. 1). Hence, adversarial training can be understood as a powerful data augmentation using hard model-specific samples. Though it is not the main focus of this work, in § 6, we show that adversarial training can be used to improve the robustness and generalization of a neural combinatorial solver.

**Remarks on decision problems.** For the binary decision problems, we typically are not required to know $\tilde{Y}$; it suffices to know $\tilde{y}$. Moreover, for such binary problems, we keep the solution constant $y = \tilde{y}$, but there also exist practical perturbations that change the label of the decision problem. For example for SAT, we can add a set of clauses that are false in isolation which makes $\tilde{y} = 0$.

**Requirements for neural solvers.** We study neural combinatorial solvers $f_\theta$ that are often a Graph Neural Network (GNN). We then solve Eq. 1 using different variants of Projected Gradient Descent (PGD) and therefore assume the model to be differentiable w.r.t. its inputs (see § D). For non-differentiable models, one can use derivative-free optimization (Yang & Long, 2021).

## 4 SAT

We first introduce the problem as well as notation and then propose the perturbation models (§ 4.1). Last, we discuss the attacks for a specific neural solver (§ 4.2).

**Problem statement.** The goal is to determine if a boolean expression, e.g. $(v_1 \vee v_2 \vee \neg v_3) \wedge (v_1 \vee v_3)$, is satisfiable $y = 1$ or not $y = 0$. Here, we represent the boolean expressions in Conjunctive Normal Form (CNF) that is a conjunction of multiple clauses $k(v_1, \ldots, v_n)$ and each clause is a disjunction of literals $l_i$. Each literal is a potentially negated boolean variable $l_i \in \{\neg v_i, v_i\}$ and w.l.o.g. we assume that a clause may not contain the same variable multiple times. In our standard notation, a problem instance $\boldsymbol{x}$ represents such a boolean expression in CNF. A solution $Y \in \{l_1^*, \ldots, l_n^* \mid l_i^* \in \{\neg v_i, v_i\}\}$ provides truth assignments for every variable. Hence, in the example above $\boldsymbol{x} = (v_1 \vee v_2 \vee \neg v_3) \wedge (v_1 \vee v_3)$, $y = 1$, and a possible solution is $Y = \{v_1, \neg v_2, v_3\}$. Note that multiple optimal $Y$ exist but for our attacks it suffices to know one.

### 4.1 Sound Perturbation Model

We now introduce a *sound and efficient* perturbation model for SAT which we then use for an adversarial attack on a neural (decision) SAT solver. Recall that the perturbation model is sound since we provably obtain the correct label $\tilde{y}$ and it is efficient since we achieve this without using a solver. Instead of using a solver, we leverage invariances of the SAT problem.

**Proposition 1** *Let $\boldsymbol{x} = k_1(v_1, \ldots, v_n) \wedge \ldots k_m(v_1, \ldots, v_n)$ be a boolean statement in Conjunctive Normal Form (CNF) with $m$ clauses and $n$ variables. Then $\tilde{\boldsymbol{x}}$, a perturbed version of $\boldsymbol{x}$, has the same label $y = \tilde{y}$ in the following cases:*

- *SAT: $\boldsymbol{x}$ is satisfiable $y = 1$ with truth assignment $Y$. Then, we can arbitrarily remove or add literals in $\boldsymbol{x}$ to obtain $\tilde{\boldsymbol{x}}$, as long as one literal in $Y$ remains in each clause.*

- *DEL: $\boldsymbol{x}$ is unsatisfiable $y = 0$. Then, we can obtain $\tilde{\boldsymbol{x}}$ from $\boldsymbol{x}$ through arbitrary removals of literals, as long as one literal per clause remains.*

- *ADC: $\boldsymbol{x}$ is unsatisfiable $y = 0$. Then, we can arbitrarily remove, add, or modify clauses in $\boldsymbol{x}$ to obtain $\tilde{\boldsymbol{x}}$, as long as there remains a subset of clauses that is unsatisfiable in isolation.*

### 4.2 Neural SAT Solver

Selsam et al. (2019) propose NeuroSAT, a neural solver for satisfiability-problems that uses a message-passing architecture (Gilmer et al., 2017) on the graph representation of the boolean expressions. The SAT problem is converted into a bipartite graph consisting of clause nodes and literal nodes. For each variable there exist two literal nodes; one represents the variable and the other its negation. If a literal is contained in a clause its node is connected to the respective clause node. NeuroSAT then recursively updates the node embeddings over the message-passing steps using this graph, and in the last step, the final vote $\hat{y} \in \{0, 1\}$ is aggregated over the literal nodes.

**Attacks.** We then use these insights to craft perturbed problem instances $\tilde{\boldsymbol{x}}$ guided by the maximization of the loss $\ell_{adv_G}$ (see Eq. 1). Specifically, for **SAT** and **DEL**, two of admissible perturbations defined in Proposition 1, we optimize over a subset of edges connecting the literal and clause nodes where we set the budget $\Delta$ relatively to the number of literals/edges in $\boldsymbol{x}$. For **ADC** we additionally concatenate $d$ additional clauses and optimize over their edges obeying $\Delta$ but keep the remaining $\boldsymbol{x}$ constant. If not reported separately, we decide for either **DEL** and **ADC** randomly with equal odds.

**$L^0$-PGD.** The addition and removal of a limited number of literals is essentially a perturbation with budget $\Delta$ over a set of discrete edges connecting literals and clauses. Similarly to the $L^0$-PGD attack of Xu et al. (2019), we continuously relax the edges in $\{0, 1\}$ to $[0, 1]$ during optimization. We then determine the edge weights via projected gradient descent s.t. the weights are within $[0, 1]$ and that we obey the budget $\Delta$. After the attack, we use these weights to sample the discrete perturbations in $\{0, 1\}$. In other words, the attack continuously/softly adds as well as removes literals from $x$ during the attack and afterward we sample the discrete perturbations to obtain $\tilde{x}$. For additional details about the attacks, we refer to § E.

## 5 TSP

We first introduce the TSP including the necessary notation. Then, we propose a perturbation that adds new nodes s.t. we know the optimal route afterward (§ 5.1). In § 5.2, we detail the attack for a neural decision TSP solver and, in § 5.3, we describe the attack for a model that predicts the optimal TSP route $Y$.

**Problem statement.** We are given a weighted graph $\mathcal{G} = (\mathbb{V}, \mathbb{M})$ that consist of a finite set of nodes $\mathbb{V}$ as well as edges $\mathbb{M} \subseteq \mathbb{V}^2$ and a weight $\omega(e)$ for each possible edge $e \in \mathbb{V}^2$. We use the elements in $\mathbb{V}$ as indices or nodes interchangeably. The goal is then to find a permutation $\sigma$ of the nodes $\mathbb{V}$ s.t. the cost of traversing all nodes exactly once is minimized (i.e. the Hamiltonian path of lowest cost):

$$\sigma^* = \arg \min_{\sigma' \in \mathbb{S}} c(\sigma', \mathcal{G}) = \arg \min_{\sigma' \in \mathbb{S}} \omega(\sigma'_1(\mathbb{V}), \sigma'_n(\mathbb{V})) + \sum_{i=1}^{n-1} \omega(\sigma'_i(\mathbb{V}), \sigma'_{i+1}(\mathbb{V})) \qquad (2)$$

where $\mathbb{S}$ is the set of all permutations and $n = |\mathbb{V}|$ is the number of nodes. Although multiple $\sigma^*$ might exist here it suffices to know one. An important special case is the "metric TSP", where the nodes represent coordinates in a space that obeys the triangle inequality (e.g. euclidean distance). For notational ease, we interchangeably use $\sigma$ as a permutation or the equivalent list of nodes. Moreover, we say $\sigma$ contains edge $(I, J) \in |\mathbb{M}|$ if $I$ and $J$ are consecutive or the first and last element. In our standard notation $x = \mathbb{G}$, $Y = \sigma^*$, and the respective decision problem solves the question if there exist $c(\sigma^*) \leq c_0$ of at most $c_0$ cost.

### 5.1 SOUND PERTURBATION MODEL

Adversarially perturbing the TSP such that we know the resulting solution seems more challenging than SAT. However, assuming we would know the optimal route $\sigma^*$ for graph $x = \mathcal{G}$, then under certain conditions we can add new nodes s.t. we are guaranteed to know the perturbed optimal route $\tilde{\sigma}^*$. Note that this does not imply that we are able to solve the TSP in sub-exponential time in the worst case. We solely derive an efficient special case through leveraging the properties of the TSP.

**Proposition 2** *Let $\sigma^*$ be the optimal route over the nodes $\mathbb{V}$ in $\mathbb{G}$, let $Z \notin \mathbb{V}$ be an additional node, and $P, Q$ are any two neighbouring nodes on $\sigma^*$. Then, the new optimal route $\tilde{\sigma}^*$ (including $Z$) is obtained from $\sigma^*$ through inserting $Z$ between $P$ and $Q$ if $\nexists (A, B) \in \mathbb{V}^2 \setminus \{(P, Q)\}$ with $A \neq B$ s.t. $\omega(A, Z) + \omega(B, Z) - \omega(A, B) \leq \omega(P, Z) + \omega(Q, Z) - \omega(P, Q)$.*

**Corollary 1** *We can add multiple nodes to $\mathbb{G}$ and obtain the optimal route $\tilde{\sigma}^*$ as long as the condition of Proposition 2 (including the other previously added nodes) is fulfilled.*

**Corollary 2** *For the metric TSP, it is sufficient if the condition of Proposition 2 holds for $(A, B) \in \mathbb{V}^2 \setminus (\{(P, Q)\} \cup \mathbb{H})$ with $A \neq B$ where $\mathbb{H}$ denotes the pairs of nodes both on the Convex Hull $\mathbb{H} \in CH(\mathbb{V})^2$ that are not a line segment of the Convex Hull.*

### 5.2 NEURAL DECISION TSP SOLVER

Prates et al. (2019) propose a GNN (called DTSP) to solve the decision variant of the TSP for an input pair $x = (\mathcal{G}, c_0)$ with graph $\mathcal{G}$ and a cost $c_0$. DTSP predicts whether there exists a Hamiltonian cycle in $\mathcal{G}$ of cost $c_0$ or less ($y = 1$ if the cycle exists).

Based on our perturbation model, we inject adversarial nodes. For the metric TSP, we determine their coordinates by maximizing the binary cross-entropy–a continuous optimization problem. This

is easy to generalize to the non-metric TSP (omitting Corollary 2), if e.g. the triangle equality does not hold or there is no "simple" cost function concerning the node's coordinates/attributes. Then, the optimization is performed over the edge weights, but depending on what the weights represent we might need to enforce further requirements.

Unfortunately, the constraint in Proposition 2 is non-convex and it is also not clear how to find a relaxation that is still sufficiently tight and can be solved in closed form. For this reason, when the constraint for a node is violated, we use vanilla gradient descent with the constraint as objective: $\omega(P, Z) + \omega(Q, Z) - \omega(P, Q) - [\min_{A,B} \omega(A, Z) + \omega(B, Z) - \omega(A, B)]$. This penalizes if a constraint is violated. We stop as soon as the node fulfills the requirement/constraint again and limit the maximum number of iterations to three. Since some adversarial nodes might still violate the constraint after this projection, we only include valid nodes in each evaluation of the neural solver. Moreover, for optimizing over multiple adversarial nodes jointly and in a vectorized implementation, we assign them an order and also consider previous nodes while evaluating the constraint. Ordering the nodes allows us to parallelize the constraint evaluation for multiple nodes, despite the sequential nature, since we can ignore the subsequent nodes.

### 5.3 NEURAL TSP SOLVER

Joshi et al. (2019) propose a Graph Convolutional Network (ConvTSP) to predict which edges of the euclidean TSP graph are present in the optimal route. The probability map over the edges is then decoded into a permutation over the nodes via a greedy search or beam search. We use the same attack as for the TSP decision problem (see § 5.2) with the exception of having a different objective with changing label $\tilde{Y}$. Although the optimality gap $\ell(\hat{Y}, Y) = {c(\hat{Y}) - c(Y)}/{c(Y)}$ is a natural choice and common in the TSP literature (Kool et al., 2019), it proved to be tough to backpropagate through the decoding of the final solution from the soft prediction. Hence, for ConvTSP we maximize the cross-entropy over the edges. Hence, we perturb the input s.t. the predicted route is maximally different from the optimal solution $\tilde{Y}$ and then report the optimality gap.

## 6 EMPIRICAL RESULTS

In this section, we show that the assessed SAT and TSP neural solvers are not robust w.r.t. small perturbations of the input using the sound perturbation models introduced in § 4 and 5. We first discuss SAT in § 6.1 and then TSP in § 6.2. We use the published hyperparameters by the respective works for training the models. We run the experiments for at least five randomly selected seeds. We compare the accuracy on the clean and perturbed problem instances (i.e. clean vs. adversarial accuracy). Since no directly applicable prior work exists, we compare to the random baseline that randomly selects the perturbation s.t. the budget is exhausted. Moreover, we use Adam (Kingma & Ba, 2015) and early stopping for our attacks. For further details we refer to § E and § F as well as the code https://www.daml.in.tum.de/robustness-combinatorial-solvers.

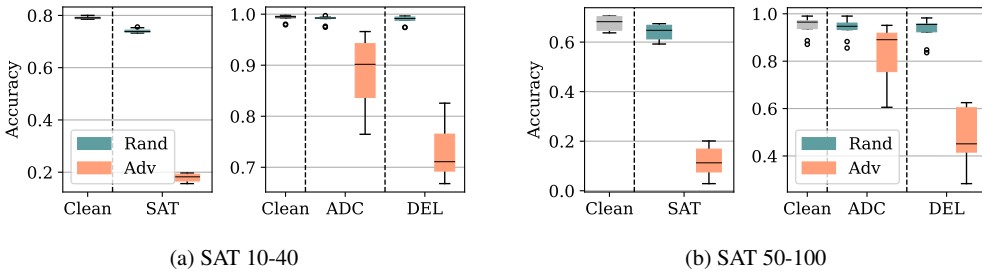

(a) SAT 10-40                    (b) SAT 50-100

Figure 2: Efficacy of adversarial attacks as introduced in § 4 and assessment of (un)robustness of the NeuroSAT model on the 10-40 dataset. Recall that SAT is the attack on the satisfiable problem instances and ADC as well as DEL are the attacks on unsatisfiable problem instances.

## 6.1 SAT

**Setup.** Following Selsam et al. (2019), we train NeuroSAT for 60 epochs using the official parameters and data generation. The random data generator for the training/validation data greedily adds clauses until the problem becomes unsatisfiable which is determined by an exact SAT solver (Ignatiev et al., 2018; Sörensson & Een, 2005). We are then left with an unsatisfiable problem instance and a satisfiable problem instance if we omit the last clause (i.e. the dataset is balanced). For each instance pair, the number of variables is sampled uniformly within a specified

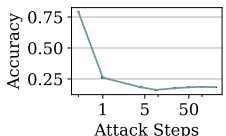

Figure 3: SAT Attack on NeuroSAT

range and then the number of literals in each clause is drawn from a geometric distribution. We name the dataset accordingly to the range of numbers of variables. For example, the training set 10-40 consists of problem instances with 10 to 40 variables. For our attacks, we use the budgets of $\Delta_{DEL} = 5\%$ as well as $\Delta_{SAT} = 5\%$ relatively to the number of literals in $x$ and for ADC we add an additional 25% of clauses and enforce the average number of literals within the new clauses.

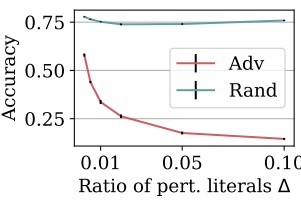

Figure 4: Rob. on satisfiable problems (SAT) over budgets $\Delta$.

**Attack efficacy and model (un)robustness.** From the results of our adversarial attacks presented in Fig. 2 it is apparent that the studied model NeuroSAT is not robust w.r.t. small perturbation of the input. Additionally, Fig. 3 shows that for the SAT attack, one gradient update step already suffices to decrease the accuracy to 26% (see also § I). All this shows the efficacy of our attacks and perturbation model, it also highlights that the standard accuracy gives a too optimistic impression of the model's performance. We hypothesize that the model likely suffers from challenges *(1) easier subproblem* and/or *(2) spurious features*, while it is also possible that the fragility is due to a lack of expressiveness.

**Difficulty imbalance.** It is much harder for the model to spot satisfiable instances than unsatisfiable ones. This is apparent from the clean accuracy and even more obvious from the adversarial accuracy. Even with moderate budgets, we are able to lower the adversarial accuracy to values below 20% for satisfiable instances while for unsatisfiable instances we barely get below 50% even on the larger problem instances 50-100. An intuitive explanation is given by the fact that it is impossible to find a solution for an unsatisfiable instance (and it is cheap to verify a candidate solution). Similarly, Selsam et al. (2019)

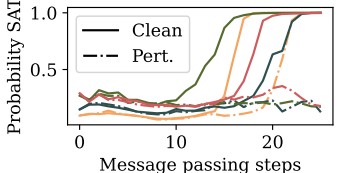

Figure 5: NeuroSAT's prediction over the message passing steps.

hypothesize that NeuroSAT only changes its prediction if it finds a solution. Thus, it is even more remarkable how we are still able to fool the neural solver in 30% of the cases (DEL attack).

**Qualitative insights.** We show in Fig. 5 how NeuroSAT's decision evolves over the message-passing steps for satisfiable instances. NeuroSAT comes to its conclusion typically after 15 message-passing steps for the clean samples. For the perturbed samples, NeuroSAT almost never comes to the right conclusion. For the instances where NeuroSAT predicts the right label, it has a hard time doing so since it converges slower. For a specific adversarial example see § J.

**Attack budget.** We study the influence of the budget $\Delta$ and, hence, the similarity of $x$ vs. $\tilde{x}$, in Fig. 4. It suffices to perturb 0.2% of the literals for the recall to drop below 50% using our SAT perturbation model. This stands in stark contrast to the random attack, where the accuracy for the satisfiable instances is almost constant over the plotted range of budgets.

**Hard model-specific samples.** The surprisingly weak performance (Fig. 2 and 4) of the random baselines shows how much more effective our attacks are and justifies their minimally larger cost (see § H). In § G, we show the difficulty is indeed

Table 1: Accuracy comparison of regular training with a 10% larger training set and adversarial finetuning of 10 extra epochs (17%).

| | Data | Regular | Extra data | Adv. train. |
|---|---|---|---|---|
| 10-40 | Train | 89.0 ± 0.06 | **89.1 ± 0.05** | 88.8 ± 0.06 |
| | Test | 89.1 ± 0.10 | 89.1 ± 0.07 | **89.6 ± 0.06** |
| | Random | 86.4 ± 0.09 | 86.3 ± 0.11 | **87.3 ± 0.07** |
| | Attack | 50.0 ± 1.16 | 49.6 ± 1.52 | **54.0 ± 0.48** |
| 50-100 | Test | 81.1 ± 0.64 | 81.2 ± 0.89 | **82.7 ± 0.50** |
| | Random | 78.6 ± 0.80 | 79.0 ± 1.02 | **80.8 ± 0.46** |
| | Attack | 39.4 ± 3.15 | 37.9 ± 2.62 | **44.4 ± 1.19** |
| | 3-10 | 92.3 ± 0.57 | 92.7 ± 0.30 | **93.0 ± 0.33** |
| | 100-300 | 64.4 ± 1.53 | 65.3 ± 1.63 | **67.0 ± 0.74** |
| | SATLIB | 66.1 ± 3.07 | **66.2 ± 4.71** | 63.7 ± 1.88 |
| | UNI3SAT | 86.0 ± 0.75 | 85.1 ± 0.80 | **86.9 ± 0.39** |

model-specific. Assuming that hard model-specific instances are the instances that are important to improve the model's performance, we can lower the amount of labeled data (potentially expensive). Of course, we cannot do anything about the NP-completeness of the problem, but adversarial robustness opens the possibility to use the expensively generated examples as effectively as possible.

**Adversarial training for SAT.** We conjecture that if the models were expressive enough, we would now be able to leverage the adversarial examples for an improved training procedure. Therefore, similar to Jeddi et al. (2020), we perform an adversarial *fine-tuning*. That is, we train the models for another 10 epochs including perturbed problem instances. We use the same setup as for the attacks presented above but we observed that too severe perturbations harm NeuroSAT's performance (e.g. a budget of 5% suffices to push the accuracy below 20% for the satisfiable instances). We therefore lower the budget of the satisfiable instances to 1% and perturb 5% of the training instances. For a fair comparison, we also compare to a model that was trained on a 10% larger training set. To compare the solvers' capability to generalize for the different training strategies, we choose datasets of different problem sizes and also include the external benchmark datasets SATLIB and UNI3SAT (Hoos & Stützle, 2000). We consistently outperform the regularly trained models in terms of robustness as well as generalization (with the exceptions SATLIB). We report the results in Table 1.

## 6.2 TSP

**Setup.** In our setup we follow Prates et al. (2019) and generate the training data by uniformly sampling $n \sim U(20, 40)$ nodes/coordinates from the 2D unit square. This is converted into a fully connected graph where the edge weights represent the $L^2$-distances between two nodes. A near-optimal solution $Y$ for training and attacks is obtained with the Concorde solver (Applegate et al., 2006). For the decision-variant of the TSP, we produce two samples from every graph: a graph with $y = 1$ and cost-query $c_0^{y=1} = c(\sigma^*) \cdot (1 + d)$ and a second graph with $y = 0$ and cost-query $c_0^{y=0} = c(\sigma^*) \cdot (1 - d)$, where $d = 2\%$ (Prates et al., 2019). For predicting the TSP solution we use ConvTSP (Joshi et al., 2019) but keep the setup identical to its decision equivalent. We attack these models via adding five adversarial nodes and adjust $\tilde{c}_0$ as well as $\tilde{Y}$ accordingly.

**Decision TSP Solver.** If a route of target cost exists, our attack successfully fools the neural solver in most of the cases. This low adversarial accuracy highlights again that the clean accuracy is far too optimistic and that the model likely suffers from challenges *(1) easier subproblem* and/or *(2) spurious features*. In Fig. 7, we see that the changes are indeed rather small and that the perturbations lead to practical problem instances. For further examples see § K.

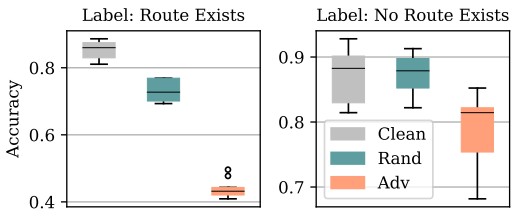

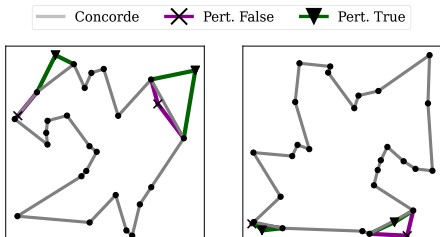

Figure 6: DecisionTSP for problems with $n \sim U(20, 40)$ nodes and five adversarial nodes.

Figure 7: Examples of the optimal route $Y$ and perturbed routes $\tilde{Y}$ for DecisionTSP.

**Difficulty imbalance.** For the TSP, we also observe an imbalance in performance between both classes. This is naturally explained with a look at the non-decision TSP version where a solver constructs a potential route $\hat{Y}$. Since $c(\hat{Y}) \geq c(Y)$ such a network comes with the optimal precision of 1 by design.

**Attacking TSP Solver.** For ConvTSP five new adversarial nodes suffice to exceed an optimality gap of 2%. Note that naive baselines such as the "farthest insertion" achieve an optimality gap of 2.3% on the clean dataset (Kool et al., 2019). Moreover, for the class "route exists" we can compare the performance on the decision TSP. Even though the model performs better than DTSP, our attacks degrade the performance relatively by 10%. In Fig. 9, we can also view the predicted routes and observe that the prediction can differ severely between the clean and perturbed problem instance.

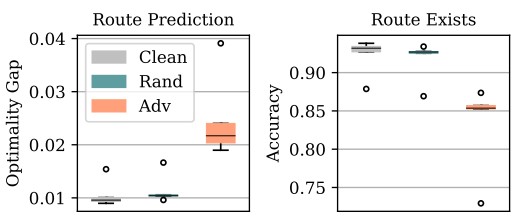

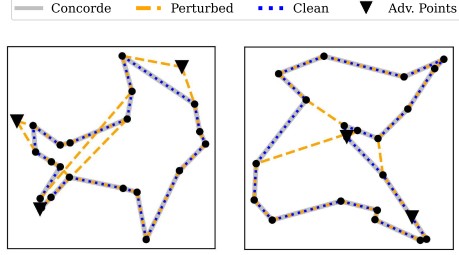

Figure 8: ConvTSP for problems with $n = 20$ nodes and five adversarial nodes. We plot the optimality gap (left) and the decision TSP performance (right).

Figure 9: Exemplary problem instances where the attack successfully changed the optimal route for ConvTSP that show drastic changes of the prediction.

## 7 RELATED WORK

**Combinatorial Optimization.** Further important works about neural SAT solvers are (Amizadeh et al., 2019; Yolcu & Póczos, 2019; Kurin et al., 2019; Cameron et al., 2020) and for neural TSP solvers (Khalil et al., 2017; Deudon et al., 2018; Bresson & Laurent, 2021; Wang et al., 2021; Bello et al., 2016; Kool et al., 2019). We refer to the recent surveys surveys (Bengio et al., 2021; Cappart et al., 2021; Vesselinova et al., 2020) for a detailed overview and discussion.

**Generalization.** There are only very few works on generalization of neural combinatorial solvers. One exception are François et al. (2019) and Joshi et al. (2021). They empirically assess the impact of different model and training pipeline design choices on the generalization (for TSP) while we discuss the generation of hard model-specific instances. A work by Selsam & Bjørner (2019) studies generalization for their NeuroSAT model (Selsam et al., 2019) and proposes a data augmentation technique relying on traditional solvers. Moreover, they study a hybrid model consisting of a simplified NeuroSAT and a traditional solver. In summary, previous works about generalization of neural combinatorial solvers analyze specific tasks while we propose a general framework for combinatorial optimization and study TSP and SAT to show its importance.

**Adversarial Robustness.** Adversarial robustness has been studied in various domains including computer vision (Szegedy et al., 2014) and graphs (Zügner et al., 2018; Dai et al., 2018). We refer to Günnemann (2021) for a broad overview of adversarial robustness of GNNs. Specifically, for TSP we optimize the continuous input coordinates (or edge weights) but use our own approach due to the non-convex constraints. For attacking the SAT model we need to perturb the discrete graph structure and rely on $L^0$-PGD by Xu et al. (2019) proposed in the context of GNNs.

**Hard sample mining.** Deriving adversarial examples might appear similar to hard sample mining (Sung, 1995). However, hard sample mining aims in spotting hard problem instances in the train data unlike we who also perturb the problem instances (of the training data or any other dataset). Moreover, if we combine augmentations with hard sample mining, we only create randomly perturbed instances and their generation is not guided by the model. The *surprisingly weak* random baseline in our experiments gives an impression about how effective such an approach might be.

## 8 DISCUSSION

We bring the study of adversarial robustness to the field of neural combinatorial optimization. In contrast to general learning tasks, we show that there exists a model with both high accuracy and robustness. A key finding of our work is that the assessed neural combinatorial solvers are all sensitive w.r.t. small perturbations of the input. For example, we can fool NeuroSAT (Selsam et al., 2019) for the overwhelming majority of instances from the training data distribution with moderate perturbations (5% of literals). We show that adversarial training can be used to improve robustness. However, strong perturbations can still fool the model, indicating a lack of expressiveness. In summary, contemporary supervised neural solvers seem to be very fragile, and adversarial robustness is an insightful research direction to reveal as well as address neural solver deficiencies for combinatorial optimization.

REPRODUCIBILITY STATEMENT

We provide the source code and configuration for the key experiments including instructions on how to generate data and train the models. All proofs are stated in the appendix with explanations and underlying assumptions. We thoroughly checked the implementation and also verified empirically that the proposed sound perturbation models hold.

ETHICS STATEMENT

Solving combinatorial optimization problems effectively and efficiently would benefit a wide range of applications. For example, it could further improve supply chain optimization or auto-routing electric circuits. Since combinatorial optimization is such a fundamental building block it is needless to detail how big of an impact this line of research could have. Unfortunately, this also includes applications with negative implications. Specifically, the examples about supply chain optimization and electric circuits also apply to military applications. Nevertheless, we believe that the positive impact can be much greater than the negative counterpart.

Unarguably, studying adversarial robustness in the context of combinatorial optimization comes with the possibility of misuse. However, not studying this topic and, therefore, being unaware of the model's robustness imposes an even greater risk. Moreover, since we study white-box attacks we leave the practitioner with a huge advantage over a possible real-world adversary that e.g. does not know the weights of the model. Aside from robustness, we did not conduct dedicated experiments on the consequences on e.g. fairness for our methods or the resulting models.

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

## A    ACCURACY ROBUSTNESS TRADEOFF FOR COMBINATORIAL SOLVERS

If the accuracy robustness trade-off existed for neural combinatorial optimization, it would imply that it is hopeless to strive for an accurate general-purpose neural combinatorial solver. Fortunately, this is not the case. To show this we built upon the ideas of Suggala et al. (2019) but we only consider the special case of combinatorial decision problems where each possible prediction $\hat{Y}$ is either true or false (i.e. no stochasticity of the true label as long as we account for the ambiguity in $Y$).

For general learning tasks, Suggala et al. (2019) refined the classical definition of adversarial robustness s.t. there is provably no such trade-off in general classification tasks if either the dataset is "margin separable" or when we restrict the perturbation space to be label-preserving for a Bayes optimal classifier.

For combinatorial optimization with a sound perturbation model $G$ we can argue both ways. First, due to the soundness we never encourage a wrong prediction (i.e. the margin condition is naturally fulfilled). This is contrast to e.g. image classification where an instance close to the decision boundary with the common $L^p$ perturbation model $\|\boldsymbol{x} - \tilde{\boldsymbol{x}}\|_p \leq r$ changes its true label or becomes meaningless (e.g. a gray image) for a large enough $r$. Second, the Bayes optimal classifier for combinatorial optimization solves the optimization problem perfectly on $p(\boldsymbol{x})$ (with support$(p(\boldsymbol{x})) \subseteq \mathbb{X}$). For these reasons, we do not need to include the Bayes optimal classifier in the definition of the attack (Eq. 1).

More formally, we show by contradiction that any minimizer $f^* = \arg\min_{f \in \mathbb{F}} R_{\text{adv},G}(f)$ is optimal w.r.t. $\min_{f \in \mathbb{F}} R(f)$. Let $\hat{f}$ be an optimal classifier w.r.t. $R(f)$ for a combinatorial decision problem.

We assume the loss $\ell$ is equal for all possible optimal $Y$, to account for the multiple possible $Y$ and define the adversarial risk

$$R_{\text{adv},G}(f) = \mathbb{E}_{\boldsymbol{x} \sim p(\boldsymbol{x})} \left[ \max_{\tilde{\boldsymbol{x}} \in G(\boldsymbol{x},Y)} \ell(f_\theta(\tilde{\boldsymbol{x}}), h(\tilde{\boldsymbol{x}}, \boldsymbol{x}, Y)) \right]$$

for any data distribution $p(\boldsymbol{x})$ and the standard risk $R(f) = \mathbb{E}_{\boldsymbol{x} \sim p(\boldsymbol{x})} [\ell(f_\theta(\boldsymbol{x}), Y)]$. W.l.o.g. we assume the best possible risk is zero $R^* = 0$ (i.e. the prediction is always true).

Suppose $f^*$ and $\hat{f}$ differ in their prediction $f^*(\boldsymbol{x}) \neq \hat{f}(\boldsymbol{x})$ over a non-empty subset of support$(p(\boldsymbol{x}))$.

Since $f^*$ is optimal and there is no stochasticity in the true label $Y$, it is always correct with standard risk $R(f^*) = 0$ and, analogously, $R_{\text{adv},G}(f^*) = 0$.

Thus, $\hat{f}$ cannot be an optimizer of $\hat{f} = \arg\min_{f \in \mathbb{F}} R(f)$ if its predictions differs over a non-empty subset of support$(p(\boldsymbol{x}))$.

Note that this also holds for support$(p(\boldsymbol{x})) \subseteq$ support$(G(p(\boldsymbol{x}))) \subseteq \mathbb{X}$ which is indeed the interesting case for contemporary data generators and models.

## B    PROOF OF PROPOSITION 1

**Proposition 1** *Let $\boldsymbol{x} = k_1(v_1, \ldots, v_n) \wedge \ldots k_m(v_1, \ldots, v_n)$ be a boolean statement in Conjunctive Normal Form (CNF) with $m$ clauses and $n$ variables. Then $\tilde{\boldsymbol{x}}$, a perturbed version of $\boldsymbol{x}$, has the same label $y = \tilde{y}$ in the following cases:*

- **SAT:** *$\boldsymbol{x}$ is satisfiable $y = 1$ with truth assignment $Y$. Then, we can arbitrarily remove or add literals in $\boldsymbol{x}$ to obtain $\tilde{\boldsymbol{x}}$, as long as one literal in $Y$ remains in each clause.*

- **DEL:** *$\boldsymbol{x}$ is unsatisfiable $y = 0$ and we obtain $\tilde{\boldsymbol{x}}$ from $\boldsymbol{x}$ through arbitrary removals of literals, as long as one literal per clause remains.*

- **ADC:** *$\boldsymbol{x}$ is unsatisfiable $y = 0$. Then, we can arbitrarily remove, add, or modify clauses in $\boldsymbol{x}$ to obtain $\tilde{\boldsymbol{x}}$, as long as there remains a subset of clauses that is unsatisfiable in isolation.*

W.l.o.g. we assume that an empty clause is true. That is, we evaluate the expression if the empty clauses were not there. Moreover, we assume to know one possible $Y$ (multiple might exist).

- **SAT:** Every disjunctive clause evaluates to one, if one literal is true. Since we keep at least one literal in $Y$ in each clause, every clause evaluates to one. The statement evaluates to one since the conjunction of true statement is also true.

- **DEL:** Since every clause is a disjunction of literals, the removal of one/some of the literals strictly reduces the number of specifiable assignments. That is, removing any literal $l_i$ from its clause removes the possibility to satisfy this clause through $l_i = 1$.

- **ADC:** Since the clauses are a conjunctive, all clauses need to evaluate to true. If a subset of clauses it not satisfiable by themselves (i.e. not all can be true at the same time), the expression necessarily resolves to 0.

Hence, all conditions in Proposition 1 do not change the satisfiability $y$, but might alter $Y$. It is apparent that the updated $Y$ can be obtained efficiently. $\square$

## C   PROOF OF PROPOSITION 2

**Proposition 2** *Let $\sigma^*$ be the optimal route over the nodes $\mathbb{V}$ in $\mathbb{G}$, let $Z \notin \mathbb{V}$ be an additional node, and $P, Q$ are any two neighbouring nodes on $\sigma^*$. Then, the new optimal route $\tilde{\sigma}^*$ (including $Z$) is obtained from $\sigma^*$ through inserting $Z$ between $P$ and $Q$ if $\nexists (A, B) \in \mathbb{V}^2 \setminus \{(P, Q)\}$ with $A \neq B$ s.t. $\omega(A, Z) + \omega(B, Z) - \omega(A, B) \leq \omega(P, Z) + \omega(Q, Z) - \omega(P, Q)$.*

**Corollary 1** *We can add multiple nodes to $\mathbb{G}$ and obtain the optimal route $\tilde{\sigma}^*$ as long as the condition of Proposition 2 (including the other previously added nodes) is fulfilled.*

**Corollary 2** *For the metric TSP, it is sufficient if the condition of Proposition 2 holds for $(A, B) \in \mathbb{V}^2 \setminus (\{(P, Q)\} \cup \mathbb{H})$ with $A \neq B$ where $\mathbb{H}$ denotes the pairs of nodes both on the Convex Hull $\mathbb{H} \in CH(\mathbb{V})^2$ that are not a line segment of the Convex Hull.*

**Proof.** We proof by contradiction. We define $(R, S) \in \mathbb{V}^2 \setminus \{(P, Q)\}$ to be the two neighboring nodes of $Z$ on $\tilde{\sigma}^*$. Suppose $\omega(P, Z) + \omega(Q, Z) - \omega(P, Q) < \omega(R, Z) + \omega(S, Z) - \omega(R, S)$ and $\tilde{\sigma}^*$ would not contain the edges $\omega(P, Z)$ as well as $\omega(Q, Z)$.

We know by optimality of $\sigma^*$ that

$$c(\tilde{\sigma}^*) - \omega(R, Z) - \omega(S, Z) + \omega(R, S) \geq c(\sigma^*)$$

and by optimality of $\tilde{\sigma}^*$ that

$$c(\sigma^*) + \omega(P, Z) + \omega(Q, Z) - \omega(P, Q) \geq c(\tilde{\sigma}^*).$$

Thus,

$$c(\sigma^*) + \omega(P, Z) + \omega(Q, Z) - \omega(P, Q) \geq c(\tilde{\sigma}^*) \geq c(\sigma^*) + \omega(R, Z) + \omega(S, Z) - \omega(R, S)$$

and equivalently

$$\omega(P, Z) + \omega(Q, Z) - \omega(P, Q) \geq \omega(R, Z) + \omega(S, Z) - \omega(R, S)$$

which leads to a contradiction.

Since we do not know what edges are contained in $\tilde{\sigma}^*$ (i.e. what nodes could be $R$ and $S$) we state the stricter condition $\nexists (A, B) \in \mathbb{V}^2 \setminus \{(P, Q)\}$ with $A \neq B$ s.t. $\omega(A, Z) + \omega(B, Z) - \omega(A, B) \leq \omega(P, Z) + \omega(Q, Z) - \omega(P, Q)$. $\square$

If multiple $\sigma^*$ exist, then this statement holds for any optimal route that has a direct connection between $P$ and $Q$. Corollary 1 follows by induction and Corollary 2 is due to the fact that the in metric space the optimal route $\tilde{\sigma}^*$ must be a simple polygon (i.e. no crossings are allowed). This was first stated for an euclidean space as "the intersection theorem" by Cutler (1980) and is a direct consequence of the triangle inequality. Also note, alternatively to the proof presented here, one can also unfold the dynamic program proposed by Rote (1991) to end up at Proposition 2.

## D  PROJECTED GRADIENT DESCENT (PGD)

**Algorithm D.1:** Projected Gradient Descent

---

**Data:** Problem $(\boldsymbol{x}, Y)$ and possibly $y$, Solver $f_\theta(\cdot)$,
   Loss $\ell$, budget $\Delta$, attack steps $s$, learn rate $\alpha$

1   $\tilde{\boldsymbol{x}}_0 \leftarrow \text{initialize}(\boldsymbol{x}, Y, \Delta)$
2   **for** $t \in \{0, 1, \ldots, s-1\}$ **do**
3    $\tilde{\boldsymbol{x}}'_{t+1} \leftarrow \text{update}(\tilde{\boldsymbol{x}}_t, \alpha_t, \nabla\ell(f_\theta(\tilde{\boldsymbol{x}}_t), h(\tilde{\boldsymbol{x}}, \boldsymbol{x}, Y)))$
4    $\tilde{\boldsymbol{x}}_{t+1} \leftarrow \text{project}(\tilde{\boldsymbol{x}}'_{t+1}, \boldsymbol{x}, \Delta)$
5   **end**
6   $\tilde{\boldsymbol{x}} \leftarrow \text{postprocess}(\tilde{\boldsymbol{x}}_E, \Delta)$
7   **return** $\tilde{\boldsymbol{x}}, h(\tilde{\boldsymbol{x}}, \boldsymbol{x}, Y)$

---

PGD is one of the most successful and widely studied approaches to craft adversarial examples. For a further techniques and a broader overview of adversarial robustness in various domains, we refer to Xu et al. (2020).

All our attacks roughly match the framework of Algorithm D.1. First, we initialize the perturbed instance, or, alternatively, we can use some variable that models the difference to the clean instance $\boldsymbol{x}$ (line 1). Initialization strategies that we consider are random initialization or initializing to the clean instance. Then we perform the attack for $s$ steps and in each step update the perturbed instance through a gradient descent step (line 3). For faster convergence we additionally use Adam as optimizer (Kingma & Ba, 2015). After the gradient update we perform a projection step that ensures we stay within the budget $\Delta$ or satisfy other constraints. For simplicity, we omit the fact that we use early stopping in all our algorithms. Specifically, in each attack step we check if the current perturbed instance $\tilde{\boldsymbol{x}}_{t+1}$ comes with the best loss so far. Then, after $s$ attack steps we assign the best possible $\tilde{\boldsymbol{x}}_s \leftarrow \arg\max_{t\in\{1,\ldots,s\}} \ell(f_\theta(\tilde{\boldsymbol{x}}_t), h(\tilde{\boldsymbol{x}}_t, \boldsymbol{x}, Y))$. This happens right before we (optionally) perform a postprocessing. Finally, we return the perturbed instance $\tilde{\boldsymbol{x}}$ with solution $Y$ or decision label $y$.

**Limitations.** As discussed in § 3, we require the model to be differentiable w.r.t. its input. Fortunately, most neural combinatorial solvers rely on GNNs and therefore this does not impose an issue. However, even if assessing a non-differentiable model one could revert to derivative-free optimization (Yang & Long, 2021).

For some combinatorial optimization problems the number of variables we need to optimize over can be very large. For example, when attacking a Maximum Independent Set neural solver (Li et al., 2018) using our perturbation models for SAT, we need to construct a graph that blows up quickly. Nevertheless, this could be done with derivative-free optimization (Yang & Long, 2021) or a scalable variant of of $L^0$-PGD called projected randomized block coordinate descent (Geisler et al., 2021).

## E  SAT ATTACK DETAILS

**SAT model description.** Selsam et al. (2019) propose to model the input problem as a bipartite graph as described in § 4.2. We instantiate the model as described in their paper: over 26 message passing steps the GNN updates its embeddings of size 128. Clause nodes and literal nodes have separate LSTMs to update their embeddings with messages produced by 3-layer MLPs. After the last message passing steps, the output vote on whether the problem is satisfiable or not is obtained by transforming the clause embeddings with an additional vote-MLP and averaging over the final votes of the literal nodes. The paper also notes that the number of message passing steps has to be adapted to the problem size. As no specific values are provided, we use 64 steps for the 50-100 dataset and 128 steps for the SATLIB and uni3sat data.

Selsam et al. (2019) trained their model in a single epoch on a large dataset consisting of "millions of samples". However, since Selsam et al. (2019) did not publish their dataset we used a total of 60,000 samples with the very same data generation strategy. We then use 50,000 of the samples to train the model for 60 epochs. With this strategy we closely match the reported performance. For the larger train set in Table 1 we generate an additional 5,000 samples. During training we use the same hyperparameters as described in the paper. Please use the referenced code for exactly reproducing the dataset.

**Optimization Problem.** We restate the optimization problem for an adversarial attack on a SAT decision problem. The **SAT** attack tries to find a perturbed instance $\tilde{x}$ that maximizes the loss s.t. every clause contains at least one literal $l_j$ that is present in the solution assignments $Y = \{l_1^*, \ldots, l_n^* \mid l_i^* \in \{\neg v_i, v_i\}\}$. Here we omit that we additionally constrain the number of inserted/removed literals relatively to the number of literals in the clean instance $x$ via budget $\Delta$.

$$\max_{\tilde{x}} \ell(f_\theta(\tilde{x}), y = 1) \quad \text{s.t.} \quad \forall \tilde{k}_i \in \tilde{x} : (\exists l_j \in \tilde{k}_i \text{ with } l_j = l_j^*) \tag{E.1}$$

The **ADC** attack maximizes the loss by adding clauses to the problem, meaning that every clause $k_i$ in the original problem $x$ has to be present also in the perturbed instance $\tilde{x}$, as these clauses ensure that the problem remains unsatisfiable:

$$\max_{\tilde{x}} \ell(f_\theta(\tilde{x}), y = 0) \quad \text{s.t.} \quad \forall k_i \in x : k_i \in \tilde{x} \tag{E.2}$$

Lastly, the **DEL** attack optimizes over what literals to delete from the problem's clauses, as long as no clause is removed completely. This results in the following optimization problem:

$$\max_{\tilde{x}} \ell(f_\theta(\tilde{x}), y = 0) \quad \text{s.t.} \quad \forall \tilde{k}_i \in \tilde{x} : \tilde{k}_i \subseteq k_i \quad \wedge \quad \text{nonempty}(\tilde{k}_i) \tag{E.3}$$

**Attack Details.** The attack on SAT problems modifies the literals that are contained in clauses. This means specifically that we optimize over the edges represented by the literals-clauses adjacency matrix $x = A \in \{0, 1\}^{2n \times m}$. We implemented these attacks such that they can operate on batches of problems, however we omit this at this point in the following for an improved readability.

Algorithm E.1 describes in detail **SAT** and **DEL** (see Proposition 1). The difference of the adjacency matrix $A$ to the adversarially perturbed version $\tilde{A}$ is modelled via the perturbation matrix $M$: ($\tilde{x} = \tilde{A} = A \oplus M$). We then optimize over $M$. During the **SAT** attack, we allow deletions and additions of edges, as long as one solution-preserving truth assignment per clause, described by the indicator $T = \text{onehot}(Y)$, is preserved (line 7). For the **DEL** attack only deletions are allowed, under the constraint that no clause can be fully deleted (lines 9 & 10). Additionally, a global budget is enforced (line 6). For details on the budget as well as other hyperparameters of the attack, we refer to Table E.1. For the **ADC** attack described in Algorithm E.2, the perturbation matrix $M \in \{0, 1\}^{2n \times \tilde{m}}$ describes additional clauses appended to the original problem (line 4). The attack can freely optimize over $M$ but the number of literals/edges. $\Delta$ is enforced s.t. on average each clause contains as many literals as a clause in $A$ (line 6).

Table E.1: Hyperparameters for the attacks on NeuroSAT proposed in § 4

|  | SAT | DEL | ADC |
| --- | --- | --- | --- |
| attack steps | 500 | 500 | 500 |
| learning rate | 0.1 | 0.1 | 0.1 |
| fraction of perturbed literals $\Delta$ | 5% of edges | 5% of edges | 25% of clauses |
| # final samples | 20 | 20 | 20 |
| temperature scaling | 5 | 5 | 5 |

Because the attack optimizes over a set of discrete edges with a gradient based method, similarly to (Xu et al., 2019), we relax the edge weights to $[0, 1]$ during the attack. Before generating the perturbed problem instance $\tilde{A}$, we sample the discrete $M'$ from a Bernoulli distribution where the entries of the matrix $M$ represent the probability of success. In contrast to (Xu et al., 2019) we sample 19 instead of 20 times but add an additional sample that chooses the top elements in $M$. Thereafter, we take the sample that maximizes the loss. Moreover, our projection differs slightly from the one proposed by Xu et al. (2019) since we iteratively enforce the budget instead of performing a bisection search.

**Algorithm E.1:** SAT & DEL Attack

**Data:** Adjacency $\boldsymbol{A} \in \{0,1\}^{2n \times m}$, edge budget $\Delta$, steps $s$, learning rate $\alpha$, solution $\boldsymbol{T} \in \{0,1\}^{2n \times m}$, SAT model $f_\theta$, label $y$

**Result:** Perturbed Adjacency $\tilde{\boldsymbol{A}}$

1 Initialize $\boldsymbol{M} \leftarrow \boldsymbol{0}^{2n \times m}$,
2 **for** $t \in \{0, 1, \ldots, s-1\}$ **do**
3     **if** $y$ **then** $\tilde{\boldsymbol{A}} \leftarrow \boldsymbol{A} \oplus \boldsymbol{M}$
4     **else** $\tilde{\boldsymbol{A}} \leftarrow \boldsymbol{A} - \boldsymbol{M}$
5     $\boldsymbol{M} \leftarrow \text{update}(\boldsymbol{M}, \alpha, \nabla \ell(f_\theta(\tilde{\boldsymbol{A}}), y)$
6     $\boldsymbol{M} \leftarrow \text{project-budget}(\boldsymbol{M}, \Delta)$
7     **if** $y$ **then** $\boldsymbol{M} \leftarrow \boldsymbol{M} * \boldsymbol{T}$
8     **else**
9         $\boldsymbol{M} \leftarrow \boldsymbol{M} * \boldsymbol{A}$
10         $\boldsymbol{M} \leftarrow \text{ensure-no-del}(\boldsymbol{M})$
11     **end**
12 **end**
13 $\boldsymbol{M}' \leftarrow \text{sample}(\boldsymbol{M})$
14 $\tilde{\boldsymbol{A}} \leftarrow \boldsymbol{A} \oplus \boldsymbol{M}'$

**Algorithm E.2:** ADC Attack

**Data:** Adjacency $\boldsymbol{A} \in \{0,1\}^{2n \times m}$, clause budget $\omega$, steps $s$, learning rate $\alpha$, SAT model $f_\theta$

**Result:** Perturbed Adjacency $\tilde{\boldsymbol{A}}$

1 Initialize $\boldsymbol{M} \leftarrow \boldsymbol{0}^{2n \times \tilde{m}}$,
2 $\Delta \leftarrow \text{avg}(\boldsymbol{A}, \text{axis} = 0) * \tilde{m}$
3 **for** $t \in \{0, 1, \ldots, s-1\}$ **do**
4     $\tilde{\boldsymbol{A}} \leftarrow \text{append}(\boldsymbol{A}, \boldsymbol{M})$
5     $\boldsymbol{M} \leftarrow \text{update}(\boldsymbol{M}, \alpha, \nabla f_\theta(\tilde{\boldsymbol{A}}), y))$
6     $\boldsymbol{M} \leftarrow \text{project-budget}(\boldsymbol{M}, \Delta)$
7 **end**
8 $\boldsymbol{M}' \leftarrow \text{sample}(\boldsymbol{M})$
9 $\tilde{\boldsymbol{A}} \leftarrow \text{append}(\boldsymbol{A}, \boldsymbol{M}')$

**Computational complexity.** All three attacks operate for $s$ steps. Under the assumption that the models are linear w.r.t. the number of edges in the graph representation of the problem (for forward and backward pass), each step has a time complexity of $\mathcal{O}(mn)$. Therefore, the overall attack has a time complexity of $\mathcal{O}(nms)$. Under the same assumption for the memory requirements of $\nabla_{\boldsymbol{M}} \ell$, the total space complexity is $\mathcal{O}(nm)$.

# F TSP ATTACK DETAILS

For simplicity we only discuss the case of metric TSP. This is also what the used neural sovers are mainly intended for. The TSP attack is actually implemented in a batched fashion. However, for an improved readability we omit the details here. We refer to table Table F.1 for details on the attack hyperparameters.

**DTSP model.** The DTSP model employs a GNN to predict whether there exists a route of certain cost on an input graph (Prates et al., 2019). The model sequentially updates the embeddings of both nodes and edges that were initialized by a 3-layer MLP with the coordinates as input. After multiple message passing steps, the final prediction is then based on the aggregation of the resulting edge embeddings. We follow the training guidelines described in the paper and train on $2^{20}$ graph samples. The graphs are generated by sampling from the unit square ($\boldsymbol{x} \in [0,1]^{n \times 2}$) and solutions are obtained with the concorde solver (Applegate et al., 2006). Dual problem sets are built from each graph for training, validation and test data by increasing and decreasing the true cost by a small factor.

**ConvTSP model.** The ConvTSP model aims at predicting the optimal route $Y = \sigma$ over a given graph via a GCN (Joshi et al., 2019). It predicts a probability map over the edges indicating the likelihood of an edge being present in an optimal solution. These lay the basis for different solution decoding procedures. We employ their greedy search method, where the graph is traversed based on the edges with the highest probabilities. We follow the training procedure described in the paper as well as the data generation technique. Data is generated the same way as for the DTSP model, however the target represents binary indicators whether an edge is present in the optimal solution. During training, the binary cross entropy between the target and the probability maps over the edges is minimized, and a solution-decoding technique is only applied during inference.

**Optimization problem.** We again restate the optimization problem from Eq. 1 for the specific case of TSP. We add $\Delta$ nodes to the original problem $\boldsymbol{x} \in [0,1]^{n \times 2}$ to create a larger, perturbed problem

Table F.1: Hyperparameters for the attacks on TSP proposed in § 5

|  | DTSP | ConvTSP |
|---|---|---|
| maximum number of adv. nodes $\Delta$ | 5 | 5 |
| attack steps | 200 | 500 |
| learning rate | 0.001 | 0.01 |
| gradient project learning rate | 0.002 | 0.002 |
| gradient project steps | 3 | 3 |

instance $\boldsymbol{x} \in [0,1]^{(n+\Delta)\times 2}$ which maximizes the loss:

$$\max_{\tilde{\boldsymbol{x}}} \ell(f_\theta(\tilde{\boldsymbol{x}}), Y) \quad \text{s.t.} \quad \forall \tilde{x}_i \text{ with } i > n \text{ Proposition 2 holds} \tag{F.1}$$

**Attack details.** The input of the TSP is represented by the coordinates $\boldsymbol{x} \in [0,1]^{n\times 2}$ for the $n$ nodes. Additionally, we know the near-optimal route $Y$ obtained with the Concorde solver which we use as ground truth. During the TSP attack we add additional adversarial nodes $\boldsymbol{Z} \in [0,1]^{\Delta\times 2}$ to the input problem $\boldsymbol{x}$. The adversarial nodes are initialized by randomly sampling nodes until they fulfill the constraint from Proposition 2 (line 1). For the attack, as described in Algorithm F.1, we operate solely on the coordinates of the adversarial nodes and assume that the model converts the coordinates into a weighted graph. For the DTSP model, we additionally calculate the updated route cost $c_0$ (accordingly to $y$) and append it to the input. For simplicity, we omitted this special case in Algorithm F.1. Moreover, for the DTSP we do not pass $\tilde{Y}$ to the loss; instead, we use the decision label $y$ that is kept constant throughout the optimization. We then obtain the updated coordinates $\boldsymbol{Z}$ for the perturbed solution $\tilde{Y}$ (lines 5-6). In the project step (line 7), we only consider the coordinates in $\boldsymbol{Z}$ that violate the constraint. As discussed in § 5.2, we perform gradient descent on the constraint due to its non-convexity. We decide against optimizing the Lagrangian since this would require evaluating the neural solver $f_\theta$ and, therefore, is less efficient. For the projection, we update the adversarial node $i$ that violates the constraint with

$$\boldsymbol{Z}_i \leftarrow \boldsymbol{Z}_i - \eta \nabla_{\boldsymbol{Z}_i} \left[ \omega(P, \boldsymbol{Z}_i) + \omega(Q, \boldsymbol{Z}_i) - \omega(P, Q) - \left( \min_{A,B \in \boldsymbol{x}} \omega(A, Z) + \omega(B, Z) - \omega(A, B) \right) \right]$$

until the constrain is fulfilled again but for at most three consecutive steps.

---

**Algorithm F.1:** TSP Attack

**Data:** Node Coords $\boldsymbol{x} \in [0,1]^{n\times 2}$, steps $s$,
         learning rate $\alpha$, TSP model $f_\theta$,
         Adversarial Coords $\boldsymbol{Z} \in [0,1]^{\Delta\times 2}$

**Result:** Perturbed Node Coords
         $\tilde{\boldsymbol{x}} \in [0,1]^{(n+\Delta)\times 2}$

1   Initialize $\boldsymbol{Z} \leftarrow$ random-allowed-point()
2   **for** $t \in \{0, 1, \dots, s-1\}$ **do**
3      mask $\leftarrow$ is-constraint-fulfilled($\boldsymbol{Z}, \boldsymbol{x}$)
4      $\tilde{\boldsymbol{x}} \leftarrow$ append($\boldsymbol{x}, \boldsymbol{Z}[\text{mask}]$)
5      $\tilde{Y} \leftarrow$ update-solution($\tilde{\boldsymbol{x}}, Y$)
6      $\boldsymbol{Z} \leftarrow$ update($\boldsymbol{Z}, \alpha, \nabla \ell(f_\theta(\boldsymbol{W}), \tilde{Y})$)
7      $\boldsymbol{Z} \leftarrow$ project($\boldsymbol{Z}, \boldsymbol{x}$);
8   **end**
9   **return** $\tilde{\boldsymbol{x}}, \tilde{Y}$

---

**Computational complexity.** We again assume that the model has a linear time and space complexity. This time it is linear w.r.t. the $\mathcal{O}(n^2)$ number of elements in the distance matrix between the input coordinates. The most costly operation we add, is the check if the constraint is violated or not for each of the potentially added nodes $\Delta$. This operation has a time and space complexity of

Table G.1: Runtime in milliseconds for the Glucose and MiniSAT solver on 100 clean and perturbed problem instances from 50-100

|  | Perturbed | DEL | ADC | SAT |
|---|---|---|---|---|
| Glucose |  | **0.1897 ± 0.20** | **0.1944 ± 0.13** | 0.1315 ± 0.06 |
|  | ✓ | 0.0335 ± 0.03 | 0.0816 ± 0.11 | **0.1391 ± 0.08** |
| MiniSAT |  | **0.1351 ± 0.07** | **0.1510 ± 0.11** | 0.0980 ± 0.05 |
|  | ✓ | 0.0189 ± 0.02 | 0.0587 ± 0.09 | **0.1383 ± 0.31** |

$\mathcal{O}(n^2)$ since we need to evaluate the distances to all nodes (except the special case where both nodes are on the convex hull but are non-adjacent, see Corollary 2). However, checking the constraint has the same complexity as the neural solver. Therefore, the overall space complexity turns out to be $\mathcal{O}(\Delta n^2) = \mathcal{O}(n^2)$ and the time complexity is $\mathcal{O}(n^2 s)$ with the number of attack steps $s$.

## G    OFF-THE-SHELF SAT SOLVERS: TIME COST OF PERTURBED INSTANCES

**Comparison SAT Solvers.** We test two SAT-solver's runtime on all attacks for both clean an perturbed samples from the 50-100 dataset to better understand the difficulty of the samples for non-neural SAT solvers. Table G.1 shows the mean as well as the standard deviation of the Glucose (Audemard & Simon, 2009) as well as the MiniSAT (Sörensson & Een, 2005) solver. The results suggest that both solvers can more quickly find that a sample is unsatisfiable for perturbed samples from both the **DEL** and the **ADC** attacks. For perturbed satisfiable samples, the runtime increases compared to the clean sample for the MiniSAT solver while Glucose solver needs about the same time as for the clean sample to find a solution.

## H    CONVERGENCE NEUROSAT

Complementary to Fig. 5, we also present the convergence for 100 randomly chosen satisfiable instances from the 10-40 dataset. We see that the observations drawn from Fig. 5 also hold here: (1) rarely an instance is predicted as SAT despite the moderate budget of perturbing 5% of the literals and (2) if NeuroSAT identifies the sample as satisfiable it requires more message passing steps to do so.

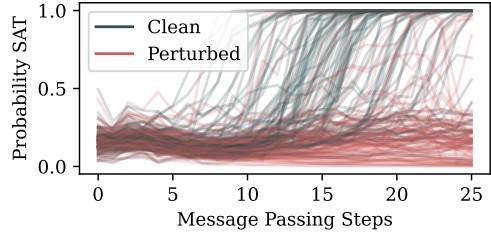

Figure H.1: Clean vs. perturbed: NeuroSAT's prediction over the message passing steps.

## I    EFFICIENCY NEUROSAT ATTACKS

To better understand the efficiency of the NeuroSAT attacks and how many gradient update steps are needed to flip the model's prediction, we show additional results for the attack strength over several attack step budgets $s$ in Fig. I.1. The indicated number of gradient update steps $s$ are taken during the attack, while retaining the learning rate from Table E.1 and early stopping on an instance level. The results in Fig. I.1 show that only 1 gradient update step suffices for the **SAT** attack to decrease the accuracy from 79% to 26%. Compared to the results from Fig. 2, where the random attack draws a single sample and only decreases the accuracy to 74%, the importance of the guidance through the gradient update becomes apparent. For the **DEL** attack on the NeuroSAT model, the attack strength

converges around 200 steps, showing again the imbalance between the labels with regards to how many misclassifications the attack can force.

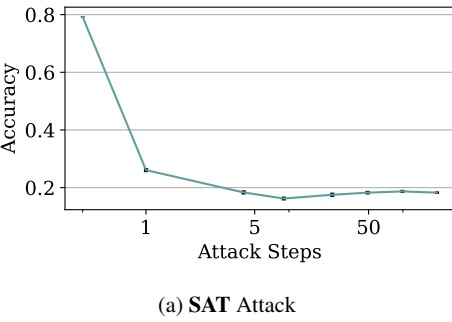

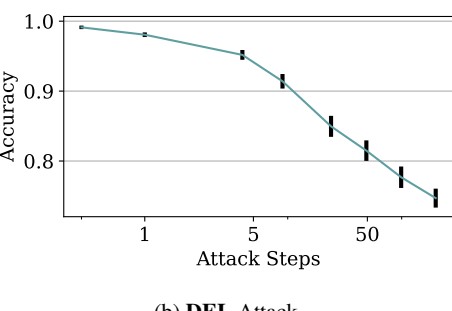

(a) **SAT** Attack                    (b) **DEL** Attack

Figure I.1: Attacks on NeuroSAT with varying number of attack steps $s$

In Table I.1 we further compare the efficiency of the attacks to the random baseline. Random samples are drawn until the loss matches or exceeds that of the adversarially perturbed sample. We introduce a cut-off at 20000 samples and report the mean number of samples for the **SAT** and **DEL** attack on the budgets $\Delta = 0.01$ and $\Delta = 0.05$.

Table I.1: Number of random samples to match optimized loss

| $\Delta =$ | 0.01 | 0.05 |
|---|---|---|
| SAT | 1257 | 9532 |
| DEL | 749 | 4854 |

We can observe that the additional computational cost of generating adversarial samples is justified, given the random baseline performs comparably well only with several thousand random samples drawn. While the efficiency of the proposed approach in general is important to consider for practical relevance, it is not sufficient to only compare to the random baseline in terms of sample efficiency. There are several other aspects to consider, like additional computational and storage overhead for a larger, denser dataset or to what extent a larger dataset of random or adversarial examples can improve generalization.

## J    QUALITATIVE RESULTS SAT

To illustrate the changes to SAT problems, we provide an example for a successful attack on a small and fairly simple problem (from SAT-3-10). While the model recognizes the problem below as satisfiable with 100% confidence, the **SAT** attack perturbes it (modified clauses highlighted blue) such that the model votes 'satisfiable' with only 0.59% confidence.

Clean SAT Problem:
$(\neg4 \vee \neg3 \vee 1 \vee 2) \wedge (1 \vee 2 \vee 3 \vee 4) \wedge (\neg4 \vee \neg3 \vee \neg2 \vee \neg1) \wedge (\neg3 \vee 2) \wedge ((\neg2 \vee \neg1 \vee 3 \vee 4) \wedge (\neg3 \vee 1 \vee 2 \vee 4) \wedge (\neg4 \vee \neg2 \vee 1 \vee 3) \wedge (\neg4 \vee \neg3 \vee \neg2 \vee \neg1) \wedge ((\neg4 \vee \neg2 \vee 1 \vee 3) \wedge (\neg2 \vee 1 \vee 4) \wedge (\neg4 \vee \neg2 \vee 1 \vee 3) \wedge (\neg3 \vee \neg2 \vee \neg1 \vee 4) \wedge (\neg4 \vee \neg1) \wedge ((\neg2 \vee \neg1 \vee 3 \vee 4) \wedge (\neg3 \vee \neg2 \vee \neg1) \wedge (1 \vee 2 \vee 3 \vee 4) \wedge (1 \vee 2 \vee 3 \vee 4) \wedge (\neg4 \vee \neg2 \vee 1) \wedge (\neg4 \vee \neg3 \vee \neg2 \vee \neg1) \wedge (1 \vee 2 \vee 3 \vee 4) \wedge (\neg3 \vee \neg1 \vee 2 \vee 4) \wedge ((\neg4 \vee \neg3 \vee \neg1 \vee 2) \wedge (\neg4 \vee 1 \vee 3) \wedge (\neg4 \vee \neg2 \vee 3) \wedge (\neg2 \vee \neg1 \vee 3 \vee 4) \wedge (\neg3 \vee \neg1 \vee 2 \vee 4) \wedge (\neg4 \vee \neg1 \vee 3) \wedge (\neg4 \vee \neg2 \vee \neg1 \vee 3) \wedge (\neg3 \vee \neg2 \vee 1) \wedge (1 \vee 2 \vee 3 \vee 4) \wedge (\neg4 \vee \neg3 \vee 2) \wedge (\neg3 \vee \neg1 \vee 4) \wedge (\neg3 \vee \neg1)$

Perturbed SAT Problem:
$(\neg4 \vee \neg3 \vee 1 \vee 2) \wedge (1 \vee 2 \vee 3 \vee 4) \wedge (\neg4 \vee \neg3 \vee \neg2 \vee \neg1) \wedge (\neg3 \vee 2) \wedge (\neg2 \vee \neg1 \vee 3 \vee 4) \wedge (\neg3 \vee 2 \vee 4) \wedge (\neg4 \vee \neg2 \vee 1 \vee 3) \wedge (\neg4 \vee \neg3 \vee \neg2 \vee \neg1) \wedge (\neg4 \vee \neg2 \vee 1 \vee 3) \wedge (\neg2 \vee 4) \wedge (\neg4 \vee \neg2 \vee 1 \vee 3) \wedge (\neg3 \vee \neg2 \vee \neg1 \vee 4) \wedge (\neg4 \vee \neg1) \wedge (\neg2 \vee \neg1 \vee 3 \vee 4) \wedge (\neg3 \vee \neg1) \wedge (1 \vee 2 \vee 3 \vee 4) \wedge (1 \vee 2 \vee 3 \vee 4) \wedge (\neg4 \vee \neg2 \vee 1) \wedge (\neg4 \vee \neg3 \vee \neg2 \vee \neg1) \wedge (1 \vee 2 \vee 3 \vee 4) \wedge (\neg3 \vee \neg1 \vee 2 \vee 4) \wedge (\neg4 \vee \neg3 \vee \neg1 \vee 2) \wedge (\neg4 \vee 3) \wedge (\neg4 \vee \neg2 \vee 3) \wedge (\neg2 \vee \neg1 \vee 3 \vee 4) \wedge (\neg3 \vee \neg1 \vee 2 \vee 4) \wedge (\neg4 \vee \neg1 \vee 3) \wedge (\neg4 \vee \neg2 \vee \neg1 \vee 3) \wedge (\neg3 \vee \neg2 \vee 1) \wedge (1 \vee 2 \vee 3 \vee 4) \wedge (\neg4 \vee \neg3 \vee 2) \wedge (\neg3 \vee \neg1 \vee 4) \wedge (\neg3 \vee \neg1)$

# K    QUALITATIVE RESULTS TSP

In this section we complement the figures Fig. 7 and Fig. 9 with further examples. In Fig. K.1, we plot further examples for the attack on DTSP and further examples for ConvTSP in Fig. K.2.

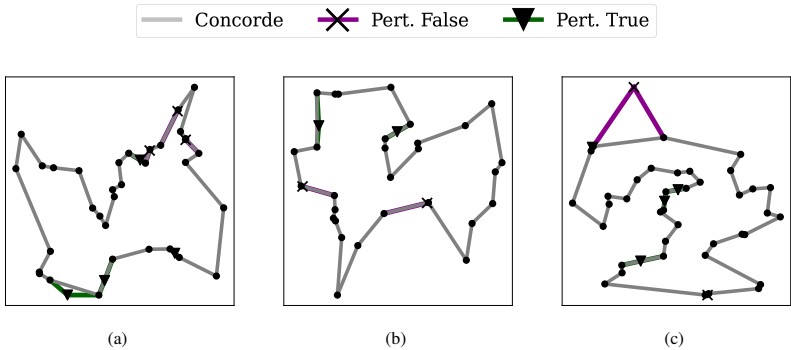

Figure K.1: Examples of the optimal route $Y$ and perturbed routes $\tilde{Y}$ for DecisionTSP.

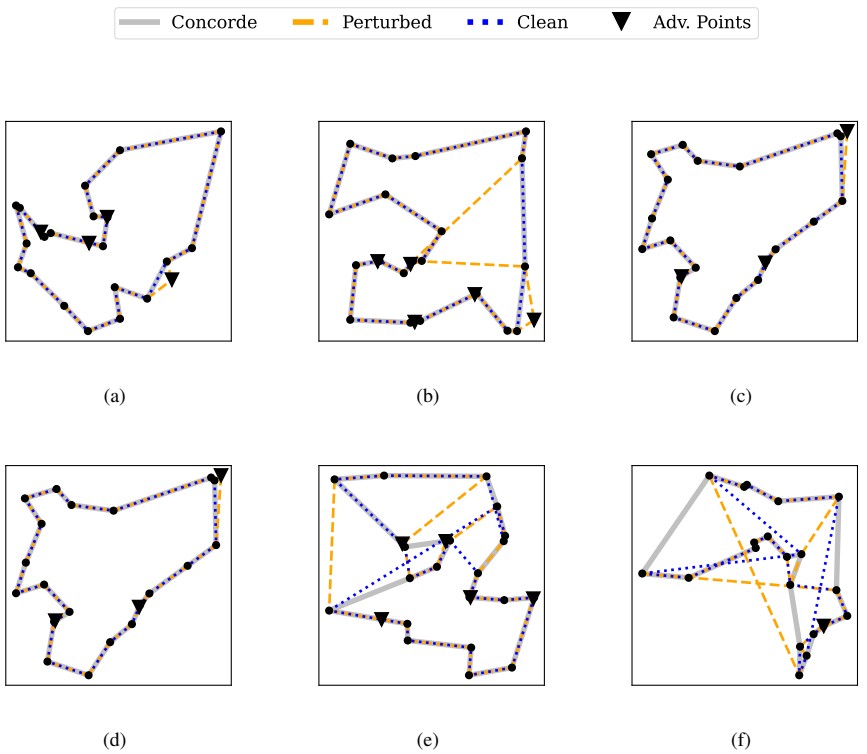

Figure K.2: Exemplary problem instances where the attack successfully changed the optimal route for ConvTSP that show drastic changes of the prediction.

