# OpenReview forum: "Generalization of Neural Combinatorial Solvers Through the Lens of Adversarial Robustness"
_ICLR.cc/2022/Conference — ICLR 2022 Poster_

### Official Review · Reviewer_7Apu · 2021-10-31

**Correctness:** 4
**Technical Novelty And Significance:** 2
**Empirical Novelty And Significance:** 2
**Recommendation:** 8
**Confidence:** 4

**Main Review:**

First, I should say that although I am no expert in the area of adversarial robustness, I am used to working in combinatorial optimisation and automated reasoning (in particular, SAT solving). Hence, I view this paper from the perspective of a SAT and CP practitioner.

Language-wise the paper seems to be well written. The discussion flow is clear and easy to follow. The notation and the concepts studied are properly introduced. Some minor issues with the notation include the lack of relation of "y" and the corresponding CNF formulas as well as the inconsistency in the use of "y = 0" and "y = -1" for unsatisfiable formulas. Another minor issue is that the example solution Y ignores
variable v3. Regarding notation, I should also say that the paper implicitly assumes what "clean accuracy" and "perturbed accuracy" are - it would help to define them explicitly.

To the best of my knowledge, all the theoretical claims presented in the form of propositions are correct. Moreover, the experimental results convincingly show that the SOTA neural solvers suffer from the lack of adversarial robustness, which is crucial given the hardness of combinatorial problems and the vital need to solve them exactly. I do believe this is a nice result that shows clear limitations of the current generation of neural combinatorial solvers and, more importantly, a possible way towards resolving the problem (more on this in the next paragraph).

It is also interesting to see that additional (adversarial) training helps alleviate the problem to some extent, although I am convinced that the issue of adversarial robustness will inevitably appear in any neural solver (as it is intrinsic to ML in general). Therefore, neural solvers will never reach the point when they could be used to replace the current complete solutions. For those who disagree, the paper proposes a way to go.

**Summary Of The Paper:**

Motivated by the emergence and fast development of the area of neural optimisation solvers having been proposed for a vast number of combinatorial problems, this paper is devoted to studying the adversarial robustness of such solvers. In particular, given the intractability of the target combinatorial problems, the paper proposes to evaluate neural solvers based on adversarial robustness. As an example, the authors aim at perturbing SAT and TSP problems and evaluating the robustness of the corresponding state-of-the-art neural solvers on the perturbed problem instances. The paper shows that the existing solvers are susceptible to adversarial attacks. Moreover, the paper claims that the issue may be alleviated if additional training of the solvers is performed on the perturbed instances.

**Summary Of The Review:**

As a result, I believe the paper offers a solid contribution that should be interesting enough for the community working in the area of neural solvers for combinatorial problems. Furthermore, I believe the message conveyed by the paper on the lack of robustness in SOTA solvers must be heard as, otherwise, neural solvers will never be able to even catch up with exact and complete solutions for the studied problems and instead will continue to be perceived as no more than a nice exercise for ML practitioners.

---

> ### Author Response · Authors · 2021-11-18
> **Author Response to Reviewer 7Apu**
>
> We thank the reviewer for the feedback, specifically, the discussion of our work from a SAT and CP practitioner's view.
>
> All suggestions made by the reviewer are incorporated into the paper: we unify the notation for 'y' in Proposition 1 (Section 4.1 and B), fix the example solution in Section 4 and add additional explanation on how the accuracy is measured.

---

### Official Review · Reviewer_Eun2 · 2021-11-01

**Correctness:** 4
**Technical Novelty And Significance:** 3
**Empirical Novelty And Significance:** 4
**Recommendation:** 8
**Confidence:** 2

**Main Review:**

At the start of the paper, the authors set out clearly the problem being addressed by the work and the motivation behind it. A nice summary of the contributions is also set out early on in the paper, though the work will be of most relevance for the specific topic of learning combinatorial solvers. In terms of presentation, the paper is self-contained with relevant background material provided, followed by concise technical work. The evaluation of the work is also sufficient.

There are occasional loaded terminologies in the texts, which makes it a bit difficult to follow (for me). But that is probably unavoidable given the limited amount of space and this specific field of research.

**Summary Of The Paper:**

The paper pinpoints the problems with neural combinatorial solvers and studies the notion of adversarial robustness for the scenario. Benefits are gained by introducing such a notion, which then allows the evaluations of neural SAT and TSP solvers.

**Summary Of The Review:**

Overall, I feel this is a worthy piece of work that is relevant, presented clearly, and accompanied by a good evaluation.

---

> ### Author Response · Authors · 2021-11-18
> **Author Response to Reviewer Eun2**
>
> We thank the reviewer for the feedback on our work and the overall idea of our submission. We would appreciate any further pointers to concrete improvements in notation to improve clarity for readers from both combinatorial optimization and adversarial robustness.

---

### Official Review · Reviewer_4nmR · 2021-11-04

**Correctness:** 3
**Technical Novelty And Significance:** 2
**Empirical Novelty And Significance:** 3
**Recommendation:** 6
**Confidence:** 3

**Main Review:**

### After Author Response
Since the authors' response and the revised submission have addressed my major concerns in:
1. how to position their work among previous works, and
2. the practical efficiency of the proposed method (in the combinatorial problem showcase)

I believe this paper is of good quality and would be happy to raise my rating from 5 to 6.

----------------------------------------------
### Strengths:
#### 1. The paper is dealing with an important and interesting problem.
#### 2. The authors illustrate their insights and method well with two combinatorial problems of significant theoretical and practical interests.
#### 3. For the SAT problem, they show adversarial training improves robustness (under the same attacks, so it is expected) and generalization (which is encouraging; though for a fair comparison, sample generation efficiency may also need to be considered, see Concern 3).
#### 4. The paper is well-written and easy to follow in general.

### Weaknesses / Concerns:
As this work seems to be a direct response to Yehuda et. al. (2020) (see e.g. Abstract and Introduction of this work), I believe it would be very helpful to compare and clarify the relation of their results.
#### 1. Data generation. Yehuda et. al. (2020) argues there is no efficient data generation procedure that is complete for these NP-hard problems. This work is no exception as otherwise it contradicts the NP-hardness assumption.
#### 2. Easier subproblem. The sound and efficient perturbation model has to be incomplete, i.e. it only covers a subset of problem instances no greater than Yehuda et. al. (2020) dictates. That said, it is possible (but not necessarily true) that with the adversarial perturbation procedure, the space covered in the training set is enlarged, and is closer to the size upper bound implied by Yehuda et. al. (2020).
#### 3. Spurious features. While adding adversarially perturbed data to the training set could potentially eliminate some spurious features, so could adding more random data, as long as random data sampling distribution covers the space of adversarial samples. Since adversarial samples require more computation to generate relative to random samples, from a practical perspective it would be helpful to compare the overall efficiency of the two training paradigms, including sample efficiency and the time required to obtain a single sample.
#### 4. Novelty. While there seems to be no previous work on adversarial training of neural combinatorial solvers, this idea falls into the "Data Augmentation" category for data generation in Yehuda et. al. (2020), with some loosely related previous work.

*Reference:*

*Gal Yehuda, Moshe Gabel, and Assaf Schuster. (2020) It’s Not What Machines Can Learn, It’s What We Cannot Teach.*

Finally -- this is not so important -- there are a few typos / word duplicates that could be spotted.

**Summary Of The Paper:**

In this paper, the authors propose to evaluate and improve the robustness and generalization of neural combinatorial solvers with adversarial examples, that is, perturbed inputs that fool the neural network to generate outputs with high loss. The authors claim that their proposal reconciles the tension between the hardness results from Yehuda et. al. (2020) and the overly-optimistic evaluation results from previous work. Furthermore, they instantiate their proposal with two classic combinatorial problems -- SAT and TSP, where adversarial examples are generated without the need to run potentially exponential-time solvers. They show that adversarial examples not only expose the fragility of common neural solvers, but can also help improve their generalization and robustness via adversarial training.

**Summary Of The Review:**

### After Author Response
1. The paper presents a novel idea -- using adversarially perturbed problem instances to evaluate the generalization performance of neural combinatorial solvers, and shows it exposes the weakness of some state-of-the-art models.
2. The paper shows that adversarial training with the adversarial data may be a useful practical method for improving neural solver's generalization performance.

-----------------------------------
### Initial Summary

While I believe the technical results presented in the paper are valid, I tend to disagree with the authors on the significance / potential limits of their approach, in particular on their response to the theoretical challenges raised by Yehuda et. al. (2020).

---

> ### Author Response · Authors · 2021-11-18
> **Author Response to Reviewer 4nmR**
>
>
> We thank the reviewer for the feedback and, particularly, the critical thoughts on the relation of our submission to the previous work of Yehuda et al.
>
> **Relation to Yehuda et al. (2020) (Response to 1. & 2.)**
>
> We want to emphasize that our submission is not intended to be a direct response to Yehuda et al., even though we do develop our motivation around the findings of Yehuda et al. Our submission discusses a more general case of data generation and for clarification, we have rephrased paragraphs two and three in the introduction as well as the caption of Figure 1 to make this clear.
>
> Importantly, the main objective of our paper is *studying the adversarial robustness of neural solvers* to better understand their generalization properties. We only investigate adversarial training as one possible strategy to improve robustness (and thus generalization) but this is not our focus. We agree that adversarial training does not circumvent the hardness of the problem. We do not claim this.
>
> **Novelty (Response to 4.)**
>
> As the reviewer correctly points out, adversarial robustness of neural combinatorial solvers has not been studied before. Our sound perturbation model allows us to efficiently evaluate the robustness of such solvers. The importance of adversarial robustness is emphasized best with our experimental results. For example, we show that for NeuroSAT a small change (5%) of a satisfiable problem statement with our sound threat model lowers the accuracy from 80% to less than 20%. These findings reveal significant shortcomings of neural solvers.
>
>
> **Sample Efficiency (Response to 3.)**
>
> We agree that sample efficiency is an important aspect for the practical relevance of adversarial examples for combinatorial optimization. However, there are different perspectives to consider: (I) the cost of obtaining a random sample vs. an adversarial example, (II) how a larger amount of random/adversarial data improves the ability to generalize, and (III) the cost of having a larger/dense dataset.
>
> For perspective (I), we now added a plot of the accuracy over the attack iterations. Obtaining an adversarial example can be very cheap. For satisfiable examples, a single attack iteration suffices to push the accuracy from 79% down to 26%. The cost of an attack iteration is here dominated by the cost of one forward and backward pass (linear in the number of neurons). Moreover, randomly generating larger problem solution pairs can be very expensive, while finding an adversarial example becomes easier for larger problem instances (see Fig. 2b).
>
> (II) In Table 2, we compare a model trained on a larger training set and a model after adversarial finetuning. We see that the adversarially trained model very consistently outperforms the model trained with more random data. Hence, we argue that it is vital to generate model-specific hard instances. Similarly, in Table I.1, we find that we need a large number of random samples to find a sample as strong as the one obtained with an attack.
>
> (III) Since the problem space is typically exponentially large w.r.t. to the problem size, it is intractable to collect a dense sample even for moderate problem sizes. We cover the space around each instance in the dataset when considering adversarial robustness and, hence, it is questionable if random samples are more effective.

---

### Decision · Program_Chairs · 2022-01-20

**Decision:**

Accept (Poster)

**Comment:**

The paper studies how neural combinatorial solvers can be susceptible to adversarial examples and what implications does this susceptibility have on the evaluation of neural solvers. Besides proposing some successful adversarial attacks, the authors provide a method for adversarial training and show its effectiveness on improving robustness and generalization. All the reviewers agreed that this paper provides a set of very interesting and novel results.